# Finding Well-Coupled Optical Fiber Locations for Railway Monitoring Using Distributed Acoustic Sensing

**DOI:** 10.3390/s23146599

**Published:** 2023-07-22

**Authors:** Felipe Muñoz, Javier Urricelqui, Marcelo A. Soto, Marco Jimenez-Rodriguez

**Affiliations:** 1Uptech Sensing SL, 31192 Mutilva Baja, Spain; 2Department of Electronic Engineering, Universidad Técnica Federico Santa María, Valparaíso 2390123, Chile

**Keywords:** distributed acoustic sensing, railway monitoring, train tracking, optical fiber coupling

## Abstract

Distributed acoustic sensors (DAS) utilize optical fibers to monitor vibrations across thousands of independent locations. However, the measured acoustic waveforms experience significant variations along the sensing fiber. These differences primarily arise from changes in coupling between the fiber and its surrounding medium as well as acoustic interferences. Here, a correlation-based method is proposed to automatically find the spatial locations of DAS where temporal waveforms are repeatable. Signal repeatability is directly associated with spatial monitoring locations with both good coupling and low acoustic interference. The DAS interrogator employed is connected to an over 30-year-old optical fiber installed alongside a railway track. Thus, the optical fiber exhibits large coupling changes and different installation types along its path. The results indicate that spatial monitoring locations with good temporal waveform repeatability can be automatically discriminated using the proposed method. The correlation between the temporal waveforms acquired at locations selected by the algorithm proved to be very high considering measurements taken for three days, the first two on consecutive days and the third one a month after the first measurement.

## 1. Introduction

Distributed optical fiber sensors [1] have emerged as one of the most effective methods for monitoring civil structures. This is primarily due to the inherent advantages of optical fibers, including electromagnetic interference immunity, corrosion resistance, electrical isolation, cost efficiency, compact size, and light weight. Furthermore, these sensors offer a significant reduction in cost per sensing point, low maintenance requirements, enhanced spatial resolution, independence from electrical supply (aside from the interrogator), and reliable measurements. These favorable characteristics allow their use in harsh environments where the use of traditional electronic sensors is inherently limited, and their maintenance and periodic revision is costly for the owner’s structure. The most important advantage of distributed optical fiber sensor technologies lies in the ability to obtain information in a distributed way throughout a monitored structure, allowing even tens of kilometers to be monitored simultaneously using a single interrogator. The high spatial resolution of distributed optical fiber sensors enables disturbances to be located over the entire length of the sensing optical fiber with high accuracy [2]. In addition, the almost ubiquitous availability of unused fiber optic cables already installed provides a very attractive opportunity to repurpose them and perform distributed monitoring with existing infrastructure without the need for new installations [3].

There is a wide range of distributed optical fiber sensors based on light scattering phenomena (Raman, Brillouin and Rayleigh) occurring in an optical fiber [4]. In recent years, distributed acoustic sensors (DAS) based on Rayleigh scattering have become one of the most widely used technologies for monitoring large structures. The high sensitivity, high dynamic range and high sampling rate enable real-time distributed vibration monitoring. Among the most popular applications of this type of sensor are intrusion detection, oil and gas pipeline monitoring, structural health monitoring, seismic monitoring, and railway track monitoring [5,6].

The extended railway infrastructure is highly suitable for DAS monitoring due to its ability to detect vibrations generated by trains crossing the tracks. This monitoring technique allows for the estimation of crucial parameters, such as the train’s position, speed, and length [6,7]. These estimations serve two critical purposes. Firstly, they confirm the absence of uncoupled wagons during train journeys, ensuring safety. Secondly, they enable efficient management of rail traffic by accurately determining train position, speed, and length, thereby minimizing congestion and optimizing operational efficiency. Moreover, they also enable the detection of potential train faults through the analysis of their characteristic acoustic signatures. For instance, flat wheels or track wear can be identified based on the distinct acoustic patterns they generate [8,9]. This capability allows for the early detection of such problems, facilitating prompt maintenance and ensuring the overall safety and efficiency of railway operations. In addition, apart from the track, third party intrusions (TPI) can be detected [10], and the overhead contact line could also be equipped with optical fibers to monitor its status [11,12].

Despite the advantages of DAS, its use in large-scale monitoring involves significant challenges. On the one hand, DAS sensitivity is highly dependent on the installation and the surroundings of the monitoring fiber [13,14]. This is because the coupling to the monitoring medium or structure determines the amount of mechanical vibration transferred to the fiber cable, so a non-uniform installation leads to an uneven sensor response. Therefore, it is very difficult to achieve constant coupling when monitoring civil infrastructures extend over tens of kilometers. The situation becomes more challenging when using optical fibers previously installed for telecommunication purposes (i.e., not intended for distributed monitoring), since the fiber coupling is likely to show significant and unpredictable changes along the route. On the other hand, the signal-to-interference-and-noise ratio of each monitored point along the fiber can be strongly affected by local acoustic interference due to environmental conditions. These difficulties result in a DAS sensor being composed of several monitored locations with unknown and variable sensitivity. Due to the large volume of data generated by this type of sensor, the manual search for reliable monitoring locations (i.e., with good coupling and low interference) becomes a prohibitively time-consuming task.

Within the existing literature, numerous algorithms have been proposed to enhance the signal to noise ratio of measurements obtained through a distributed acoustic sensor. These algorithms employ techniques such as wavelet decomposition [15], empirical mode decomposition [16], curvelet transform [17], moving average, and dynamic window moving averaging [18,19], to name a few. However, it is important to note that these methods do not adequately address the diverse coupling levels encountered in DAS sensors.

Several studies have been conducted using distributed fiber optic sensing in the railway industry [6,7,8,9,20,21,22] for applications such as determining railway condition monitoring, train position, and/or speed estimation. Some of them report the use of new installations with fibers installed for monitoring purposes [8,20], thus having uniform coupling along the monitored route, while others [21,22] use already installed fibers, emphasizingthe problem of unpredictable sensitivity changes. 

Regarding the characterization of spatial locations, in [9], the fiber locations are characterized by averaging the acoustic response along the time axis for different trains on different days obtaining a “signature” of the sensor-rail system. Although this method provides information about the energy at each fiber position, it must be interpreted with care, as not all high-energy locations indicate good monitoring positions. For example, locations affected by constant high-energy acoustic interference (e.g., in locations near roads or construction sites) are expected to show non-repeatable acoustic waveforms over time. In [21], a similar study to the one presented in this paper is carried out, where a comparison of waveforms generated by different trains at the same fiber location is made, but the waveform similarity between them is not evaluated numerically. Furthermore, fiber locations were chosen randomly and were not compared along the route of the cable. In a more general context, a method to select monitoring locations with good coupling is described in [23] to apply acoustic beamforming and source location methods to DAS measurements. However, only acoustic sources at fixed positions have been analyzed, resulting in a different characterization for each source position. 

To our knowledge, existing research has acknowledged the presence of coupling variations along the optical fiber and has proposed methods to characterize these signal levels. However, thus far, there is a lack of literature on automated techniques specifically designed to identify well-coupled fiber locations for railway monitoring applications utilizing distributed acoustic sensing.

In this work, an automated method to characterize the repeatability of the temporal waveforms caused by trains and acquired at each monitoring position of a DAS is proposed to identify locations with both good mechanical coupling and low acoustic interferences. The method benefits from two important facts: (i) the distance between the railway and the monitoring fiber is fixed, and (ii) all trains running on the track are of the same model. This allows the generation of repeatable acoustic signals that disturb the fiber along its entire length for each train crossing the track, without the need for additional experiments or installations. The proposed method is validated with measurements performed over three days: two of them on consecutive days, while the third one took place one month after the first measurement. During the acquisition stage, 71 trains were automatically detected, and the waveforms acquired at an illustrative 520 m long fiber section were compared. The results indicate that spatial measurement locations with good waveform repeatability can be automatically discriminated by the proposed method. 

The structure of this paper is as follows: In the first section, the acoustic distributed sensor employed and the field experimental conditions are presented. The procedure of two essential tasks that allow the correct interpretation of the data, namely, station identification and redundant fiber removal, are explained. Then, a method used to automatically generate a dataset is described in detail. Next, the pre-processing and the automatic train detection algorithm are explained. Finally, the method of repeatability characterization of the spatial monitoring location is explained and verified using experimental data.

## 2. DAS Measurement Dataset Generation

In this section, the procedure used to obtain the acoustic waveforms generated from multiple trains is presented. First, the DAS sensor and the parameters used to acquire the data are described. Then, the location of the experiment and the sensing optical fiber are presented. Subsequently, a procedure to find the train stations is explained, as they will be used to facilitate the calculations later. Finally, the method to automatically detect trains and to extract the acoustic signals is explained. 

### 2.1. Distributed Acoustic Sensor

In this work, the UTS-AS1000 DAS interrogator from UPTECH SENSING is used to acquire the acoustic signals. DAS measurements are obtained with a pulse width of 100 ns (corresponding to a 10 m spatial resolution), a gauge length of 10 m, and pulse repetition rate of 500 Hz, which allows measuring signals up to 250 Hz according to Nyquist theorem [24]. Although the DAS can monitor up to a maximum of 40 km of fiber with good SNR; the system has been connected to an already installed optical fiber, which exhibits high losses, and therefore only 18 km of fiber could be monitored, resulting in 18,000 independent monitoring positions (so-called acoustic channels).

### 2.2. Field Experimental Conditions

Figure 1a shows the optical fiber layout of the monitored route in this study. The sensing cable used is a 30-year-old, 24-km-long, unused communications cable already installed alongside the E2 railway line from Euskal Trenbide Sarea (black line). This railway line connects the stations of Amara, Spain and Hendaye, France (labelled ‘A’ and ‘B’, respectively). The UTS-AS1000 DAS interrogator is placed at the telecommunication room at Amara station. Due to the route of this railway track, the sensing optical fiber crosses different environments, e.g., roads, city, underground tunnels, bridges, and a harbor. Each of these environments generates different levels of acoustic interference and fiber coupling. Only one type of train runs on this track, the 4-carriage Euskotren model 900.

Figure 1b shows the one-way power losses along the fiber, obtained using an optical time-domain reflectometer. The resulting experimental power loss (blue line) is compared to the theoretical power loss of a fiber with a standard attenuation coefficient of 0.2 dB/km (orange line) [4]. The results indicate that after the 18th kilometer the power loss reaches 8.2 dB, which is equivalent to the loss obtained at the end of a 40-km-long cable with a standard attenuation coefficient. This can be attributed to the typical degradation of a 30-year-old fiber. Therefore, only the first 18 km of fiber are taken into consideration.

### 2.3. Station Identification and Redundant Fiber Section Removal

When working with previously installed fiber optic cables, the prior step is to find the correspondence between the fiber cable and the railway path. It is necessary to first locate redundant fiber sections as well as fiber sections going outside the railway path. This extra optical fiber is normally installed in case any repairs need to be made to the communication link or in case the fiber routing needs to be changed. In the context of distributed monitoring, they generate spatial discontinuities in the measured acoustic signals, negatively affecting train tracking algorithms and, therefore, they should be removed before the data analysis. This could be achieved by using calibration points or “tap tests” [21], which involve a significant amount of logistics and time to process the data. However, when working along a railway track, the fiber position can be estimated directly from the measured data, exploiting the knowledge of the route and railway stations. Furthermore, redundant fiber sections can also be identified using the data, as will be explained below, saving a great amount of time and logistics.

Station locations and redundant fiber sections can be directly found from a waterfall diagram, in which the acoustic signals of several trains are plotted as a function of time and distance. Figure 2 shows a 10-min waterfall diagram for the entire monitored distance, where the color scale represents the band-pass filtered backscattered phase level (the filtering procedure is explained later in the document). Darker diagonal lines represent moving trains, while darker vertical lines correspond to static perturbations. 

Stations can be identified by finding locations where the diagonals exhibit a jump in the vertical axis, since trains do not vibrate when stopped. To illustrate this point, the area indicated by a red square, labelled “A” in Figure 2 is shown in more detail in Figure 3a.

Figure 3a shows two trains travelling in opposite directions, represented by the two discontinuous diagonal lines in the figure. Note that there are two locations where both diagonals display sharp time shifts: the first one at 2 km, and the second one at 3.5 km (indicated by blue arrows). Since trains travelling in opposite directions exhibit time jumps at the same spatial locations, it can be concluded that these positions correspond to a station (depicted at the top of figure) rather than random locations where the train has stopped. Notice also that each station has a different background interference, which is evident from the signal level at each of the detected stations. The right-hand station has a constant and stronger level background interference, which is not present at the station on the left-hand station.

Redundant fiber sections going outside the railway track can be located by finding spatial locations where the acoustic signal emitted by a train exhibits an abrupt horizontal discontinuity. These discontinuities can be observed for trains travelling in both directions. Figure 3b shows an example of this situation. Three vertical stripes with no signal for both train directions can be recognized, indicating three redundant fiber spools placed outside the track (depicted at the top of figure). By inspecting the entire fiber, the location of the 11 stations corresponding to this train line are identified and the areas with redundant fiber are eliminated for the subsequent analysis.

### 2.4. Train Detection Algorithm and Dataset Generation

To analyze the repeatability of the acoustic waveform generated by trains, a dataset must be generated containing the signals of several trains at different spatial locations. However, the signal generated by a train corresponds to a transient signal lasting a few seconds that travels over time and position in the DAS waterfall representation [9]. Consequently, the extraction of the acoustic waveform is not straightforward. To deal with this situation, an automatic recognition method is applied to DAS measurements to estimate the average speed of the train as well as the vibration time at each single spatial location. With this information, the signal generated by the trains can be extracted and cropped from the DAS measured data. To simplify the detection method only measurements between 2 known stations are used. Furthermore, the train speed is assumed constant between stations, so that the vibration generated by a train is considered as the diagonal line observed in the DAS waterfall diagram with no time jumps. This assumption does not represent a problem in this case, since a coarse estimation of the beginning and end of the temporal signal generated by trains along the fiber is intended. For this study, only signals below 12.5 Hz are analyzed to identify low-frequency load deflections of the track [8].

Figure 4 shows a schematic of the steps utilized to process and generate the acoustic waveform dataset. First, a pre-processing stage is required to select only the frequency band of interest. A moving average in a 0.04 s window is performed to neglect higher frequencies (antialiasing filter) and subsequent data are down sampled to 25 Hz. Then, since raw DAS measurements are thermally drifted, a high-pass filter at 0.8 Hz is applied to obtain a stable signal [8].

Next, an image processing stage is applied to the pre-processed data to enhance the perturbation contour and to reduce background noise. First, a binary threshold is selected to convert the filtered DAS waterfall into a binary image. The threshold is chosen to be the 50% of the maximum root-mean-squared value of the temporal signal at a specific spatial point. Any pixel that exceeds the threshold value is given a new value of 1, otherwise, the value is set to 0. Then, morphological transformations are applied to the resulting binary image. Morphological operations apply a structuring element to an input image, creating an output image of the same size. In a morphological operation, the value of each pixel in the output image is based on a comparison of the corresponding pixel in the input image with its neighbors. Thus, exploiting the spatiotemporal information available in the measurements made by the DAS sensor. A “closing” operation followed by an “opening” operation are performed to the binary image [25]. The former has the effect of filling empty spaces within the diagonal representing the moving train, while the latter cleans the background noise of the image. Finally, a train detection stage is performed. To detect diagonal lines in the binary image, an optimization problem is solved to find the best-fit line and the width of a rectangular area surrounding the diagonal. The problem is stated as follows:$$Argmax_{k,b,W\;}\overset{N}{\underset{i=0}{\sum }}D\left[i,j\right]$$
where $j\in \left\{j>k\times i+b+W\right\}\cap^{\;}\left\{j<k\times i+b-W\right\}\cap^{\;}\{0<j<M\}$, D is the N×M data matrix containing the DAS measurements. i, j corresponds to the distance and temporal indices, respectively. k is the slope of the fitted line; b is the intercept, and W is half of the width of the rectangular area. The train velocity is calculated as 1/k and the temporal duration of the signal corresponds to 2W. An example of the processing procedure applied to a DAS waterfall between known stations is shown in Figure 5.

The raw DAS waterfall is shown in Figure 5a. The results after the pre-processing, image processing, and diagonal scanning stages are shown in Figure 5b–d, respectively. Note that, after the image processing stage, a completely isolated diagonal with no background noise is obtained for this example. Consequently, line fitting is a simple task to carry out after the applied processing. The result of the linear fitting is shown in Figure 5d. The estimated average velocity corresponds to −70 km/h and the temporal width 2W of the acoustic signal is 7.92 s. From the velocity and diagonal width, the start and end time of the temporal signal at each spatial location can be selected to crop the waveform and generate the acoustic waveform dataset. For each spatial location, the time signal corresponding to a single train is stored. Then, by detecting multiple trains, a collection of temporal signals for a specific spatial position is obtained. Note that using the proposed method, the resulting signals are slightly out of phase with each other, as the linear fit gives a coarse approximation of the start and end of each time signal. 

## 3. Results

In this section, the repeatability of the temporal waveforms acquired at each spatial monitoring location is described. Then, the acoustic waveforms obtained at specific locations selected by the algorithm are illustrated.

### 3.1. Correlation Analysis

To evaluate the repeatability of the acoustic signals, measurements are performed over three days: two of them on consecutive days, while the third one took place one month later. The study is performed on an illustrative 520 m long section of track between stations (the same region shown in Figure 5), corresponding to 1300 spatial sampling points (separated by 40 cm). Figure 6 shows the schematic of the illustrative region. The train tracks are situated within a tunnel, while the optical fiber is buried underground inside a plastic pipe covered by concrete at an unknown depth. 

To reduce the variability of the measured waveforms, only trains moving in the same direction are analyzed, so that the distance between the railway track and the optical fiber is fixed for each sensing point during the study. A total of 28 trains are detected on the first day of measurement (5-h acquisition), 11 trains on the second day (2-h acquisition) and 32 trains on the last day (6-h acquisition), resulting in a total of 71 trains.

To assess the waveform repeatability at a specific measurement location, the cross-correlation function [26] between all detected acoustic waveforms is calculated and the maximum is recorded. Note that the correlation function is used in this case, as the signals are delayed temporally from each other. This process results in a matrix of size 71 × 71 for each spatial location, containing the correlation peak amplitude of each combination of waveforms generated by the detected trains. From the cross-correlation calculation, the time lag between the signals is also obtained. Figure 6 shows two illustrative examples of correlation matrices at different spatial locations (z=4372 m and z=4468 m), where the color scale represents the peak correlation value. Figure 6a,b shows spatial locations with high and low repeatability of the temporal waveforms, respectively.

The yellow diagonal lines correspond to the maximum autocorrelation value of the acoustic signals (i.e., equal to 1), while the other correlation values are obtained from combinations between different waveforms. A comparison of the two correlation matrices reveals very different results. On the one hand, Figure 7a shows a correlation matrix where there is a high correlation ranging from 0.8 to 1.0 between many of the waveforms (indicated by the green and yellow color), corresponding to a spatial location with optimal coupling and minimal interference, since the acquired waveforms exhibit good repeatability over time. On the other hand, Figure 7b shows a correlation matrix with values not exceeding 0.5 for almost all combinations (indicated by the darker blue color), corresponding to a spatial location with low waveform repeatability. This can be explained by either poor fiber coupling, a high level of acoustic interference, or both effects combined. The non-uniformity in the correlation matrix shown in Figure 7a occurs because the temporal waveform depends on the speed of the train [20]. Thus, only signals acquired when the train is travelling at close speeds will be highly correlated. However, if the number of analyzed waveforms is sufficiently large, it is likely to find signals generated by trains travelling at similar speeds, increasing the chance of finding a correlated signal within the dataset.

### 3.2. Correlation Indicator 

Since not all channels are able to monitor acoustic waveforms, it is of the utmost importance to identify the random locations where the acoustic waveforms exhibit good repeatability over time. In this study, an indicator that summarizes the information from the correlation matrix into a single value representing each spatial monitoring location is proposed. Such an indicator is obtained by counting the number waveforms under the lower diagonal that exceeds a given correlation threshold. Note that the counting of the number of waveforms is considered and not the total number of correlations, as the purpose of the indicator is to measure how many of the analyzed waveforms find at least one similar waveform within the set. The indicator is designed in this way to reduce the impact of working with a dataset containing signals generated by trains travelling at different speeds. Regardless of whether there are many different speeds, if a waveform is repeated at least once, the spatial location score will increase. The maximum possible count equals the total number of signals analyzed, which is used to normalize the indicator. Thus, the indicator expressed as a percentage can vary between 0 and 100%, representing how many of the analyzed signals found a similar waveform in the dataset.

As an example, a correlation threshold value equal to 0.7 is chosen to guarantee good similarity between acoustic waveforms. Figure 8 shows the percentage of correlated waveforms for the illustrative 520 m-long fiber section.

For each spatial location, the correlation matrix is obtained from the signals of the 71 detected trains and evaluated using the proposed indicator. The locations z=4372 m and z=4468 m (used as an example in Figure 7) are highlighted using red and black dots, respectively. The proposed indicator assigns a higher score to the spatial location with the highest number of correlated waveforms (red dot), indicating a more reliable monitoring location than the one showing low correlation (black dot).

As can be seen, the percentage of correlated waveforms at each spatial location along the fiber changes unpredictably and does not follow a specific pattern. Note that even close measurement locations exhibit different levels of correlated waveforms, reinforcing the idea that a method must be applied to automatically find useful measuring locations. By using the proposed indicator, it is possible to discriminate the repeatability of the waveforms at each spatial measuring location, thus allowing finding the most reliable positions to perform effective measurements. From the correlated waveform percentage, trusted monitoring locations can be chosen using a user-defined confidence threshold.

### 3.3. Acoustic Waveform Comparison over Time

In this section, waveforms at the same spatial location and acquired over different days are compared. Waveforms are temporally aligned using the time delay calculated from the cross-correlation function. Spatial measurement points with a high correlation count are selected to demonstrate high repeatability over time. The following notation is used to refer to different train acoustic waveforms: $T_{z,d,m}$, where z is the spatial location, and d and *m* correspond to the days and minutes of difference with respect to a reference detected train $T_{z,0,0}$.

Figure 9 shows the acoustic waveform of two trains measured at point $Z_{A}=4614\;m\;$ on the same day, 90 min apart. Note that the vertical axis of the orange line is shifted in order to better illustrate the shape of each waveform. A high correlation value of 0.9 is obtained, demonstrating the high similarity between the two signals for both signal amplitude and waveform. At the top of the figure, a 4-carriage train is also illustrated, in which each of the bogies is numbered from 1 to 8. Note that the waveform contains 8 peaks corresponding to each of the bogies of the train. The correlation value obtained between the two waveforms is shown in the lower right corner.

Figure 10 shows the acoustic waveform of two trains measured at the same spatial location $Z_{A}$, on the same day, 30 min apart. An increase in train speed with respect to the waveforms shown in Figure 9 can be appreciated, evidenced by a shortening of the waveform duration. However, a high correlation value of 0.95 is obtained between the two waveforms. This indicates that even though the waveform is dependent on the train speed, trains with the same speed generate highly repeatable signals.

Figure 11 shows the acoustic waveform of two trains measured at the same spatial location $Z_{B}=4226\;m$ on the same day, 6 min apart. An even greater increase in speed can be seen in this case, indicated by the shorter duration of the waveform. Notice that at this train speed, the peaks corresponding to bogies 2, 3 are combined into one, and the same is true for peaks corresponding to bogies 4, 5 and 6, 7 (as depicted by the numbering at each peak). This illustrates how the sensor loses the ability to count each bogie individually for this train speed. This is due to the low pass filtering effect occurring as a consequence of integrating the phase measurement along the gauge length [8]. Nevertheless, a high correlation of 0.85 is obtained between the acoustic waveforms, strengthening the idea that trains travelling through a spatial point at the same speed have repeatable waveforms over time.

Figure 12 shows the acoustic waveform of three trains obtained on three different days. A reference train obtained on day 0, a second one obtained one day and 6 min later and a third one obtained one month and 12 min after the reference. Note that a new vertical axis (yellow) is added that is also vertically offset to better illustrate the waveform of the signals. High correlation values are obtained for both cases, corresponding to 0.96 and 0.92, respectively. This demonstrates that the measurements acquired at the spatial monitoring location selected using our method exhibit high correlation even with a month’s difference. 

Figure 13 shows the waveform of two trains acquired at the same spatial location $Z_{C}\;=\;4635\;$m, on the same day, 90 min apart. The signals exhibit a very similar signal duration and temporal location of the peaks, indicating a similar train velocity, but a noticeable difference in the signal amplitude.

The origin of this amplitude increase is unknown, but as there is only one train model running on the railway, it is suspected that it may be caused by a change in the weight of the train due to a considerable increase in the number of passengers. This result indicates that the DAS sensor can recognize changes in the amplitude of the acoustic signal generated by a train, which can be presumably associated with the weight of the train.

## 4. Discussion

Applying the proposed method systematically would allow monitoring the fiber coupling over time, which could provide valuable information about the environment surrounding the fiber. For example, for fibers buried in the ground it would be possible to study how the soil compactness changes after a day of rain or after high temperatures. In the same way, it would also make possible the assessment of the coupling of different types of fiber installations, e.g., to compare whether a fiber buried in the ground has better coupling than a fiber attached to a tunnel wall, or to compare different types of attachment methods. From such studies, a decision can be made to change or upgrade the fiber installation in key areas to allow distributed monitoring at those locations. Similarly, this information can be considered in future fiber communications installations to take advantage of distributed monitoring capabilities as well.

Note that temporal signals are analyzed here instead of spatial signals, as suggested by [20]. This approach takes advantage of the localized nature of fadings, as well as the random variation in their position along the fiber. When a train passes on a railway track, vibrations are generated, resulting in fading-free signals at certain fiber locations, while others experience fading. However, due to frequency changes in the laser and the highly dynamic nature of the environment in which the fiber is installed (i.e., is affected by environmental factors and acoustic interference), the position of fadings is constantly changing over time. Thus, by creating a sufficiently large dataset, the probability of measuring the vibration generated by the passing train along the entire fiber without experiencing fading increases. This enables the comparison of waveforms from multiple trains at virtually all locations along the fiber. When a fading occurs, the correlation score assigned to that spatial location decreases, as the acquired signals will not correlate well with others. This lack of correlation is the reason why no spatial location achieves a perfect 100% correlation score. Despite this effect, the proposed method is able to characterize the spatial locations of the fiber due to the number of detected waveforms.

## 5. Conclusions

The results indicate that even by using a 30-year-old optical fiber with some degradation that is non-optimally installed for sensing applications it is possible to extract useful information from trains crossing the railway track. The acquired signals could allow the estimation of the train position, speed, bogie count, and possibly some parameters proportional to the weight of the train. The proposed method is simple enough to be incorporated in the initial set-up of any DAS sensor for railway monitoring applications, simply by acquiring calibration data and analyzing the correlation between the acquired temporal signals. The proposed method is designed in the most general way possible, so we consider that it can be used as a standard procedure when connecting a DAS sensor to a previously installed fiber, since the problems raised here will be encountered in multiple fields of application. By knowing the reliable spatial monitoring locations, the analysis to perform train or railway condition monitoring can be limited to them and the rest can be discarded if necessary. Note that even with constant coupling along the fiber, spatial locations affected by acoustic interference will also exhibit low repeatability, so it would be useful to eliminate them in some contexts. In recent years, there has been a great deal of progress and interest in the application of pattern recognition techniques and classification algorithms applied to data acquired with DAS sensors [3,27]. It has been reported that training classification algorithms using signals acquired at spatial locations with poor coupling have reduced classification performance, as signals with poor signal-to-noise ratio are added to the analysis [28]. Therefore, the proposed method can be used to improve existing pattern recognition and classification algorithms.

Considering that many railway stations and tracks have unused installed fiber worldwide, the proposed method and the obtained results emphasize the opportunity for simple and effective train monitoring using existing infrastructure, without the need for new installations. Although the use of non-optimal installations limits the distributed monitoring of the entire route, using the proposed method, it is possible to find the fiber locations that make the best use of the current infrastructure.

## Figures and Tables

**Figure 1 sensors-23-06599-f001:**
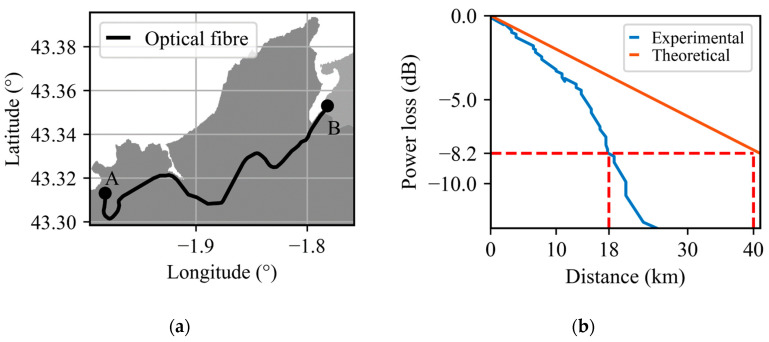
Sensing optical fiber characteristics. (**a**) 24-km-long optical fiber layout, Amara, Spain and Hendaye, France are labelled ‘A’ and ‘B’, respectively. (**b**) One-way theoretical and experimental power loss over distance.

**Figure 2 sensors-23-06599-f002:**
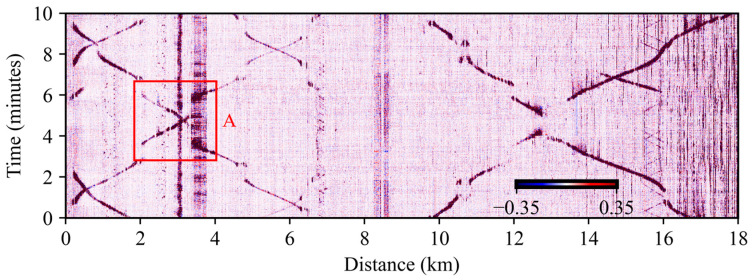
10-min waterfall diagram of DAS measurements. The red boxed zone labelled “A”, corresponds to an area of interest where trains in both directions can be seen reaching railway stations. It is discussed in more detail below.

**Figure 3 sensors-23-06599-f003:**
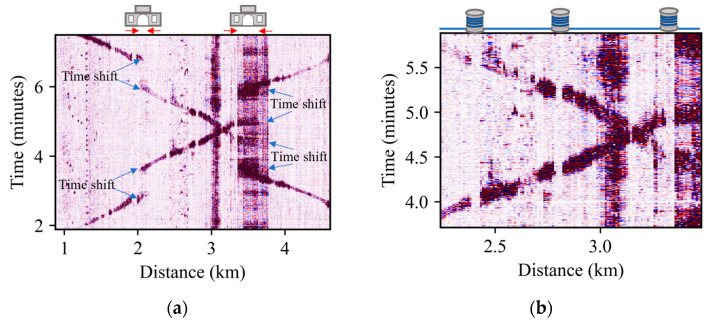
(**a**) Time shifts associated with train stations. (**b**) Signal discontinuities associated with redundant fiber spools.

**Figure 4 sensors-23-06599-f004:**
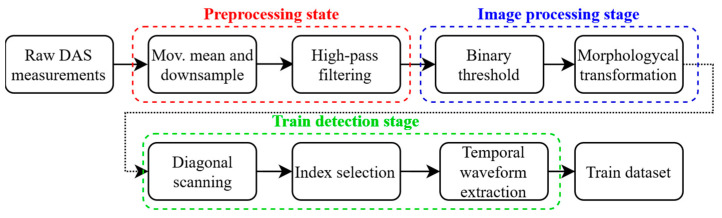
Train detection and acoustic waveform extraction scheme.

**Figure 5 sensors-23-06599-f005:**
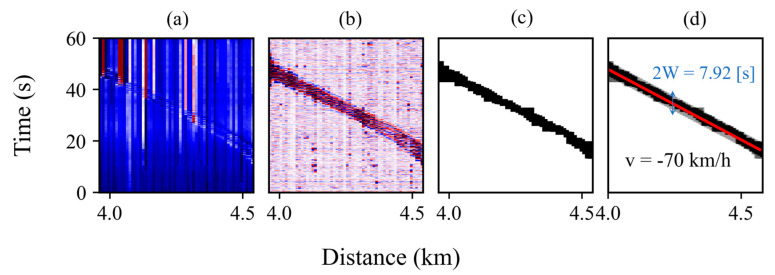
DAS waterfall data at different processing steps. (**a**) Raw DAS measurement. (**b**) Filtered DAS measurement. (**c**) Binary image. (**d**) Line-fitting results. A velocity of -70 km/h is obtained and a signal duration of 7.92 s.

**Figure 6 sensors-23-06599-f006:**
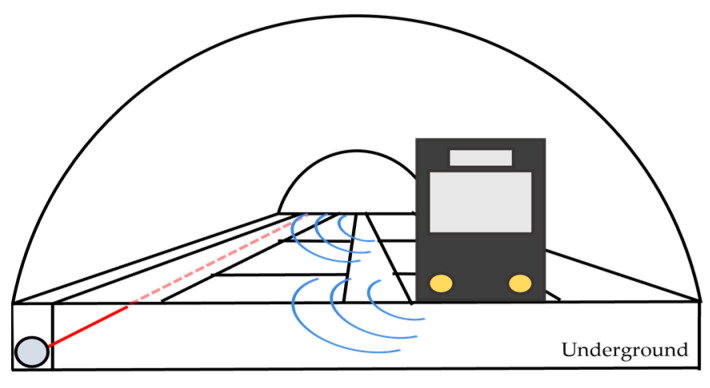
Schematic of the analyzed area. The train tracks are situated within a tunnel and a 520 manalyzed optical fiber (red line) is buried underground inside a plastic pipe covered by concrete at an unknown depth. Blue lines represent the acoustic signal generated by the train travelling to the optical fiber.

**Figure 7 sensors-23-06599-f007:**
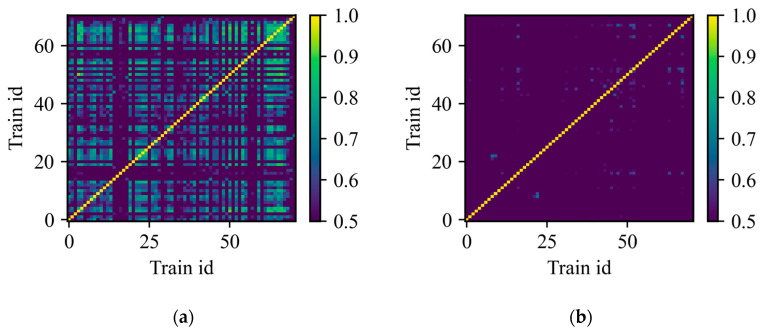
Correlation matrix for specific spatial measurement locations. (**a**) z=4372 m (**b**) z=4468 m.

**Figure 8 sensors-23-06599-f008:**
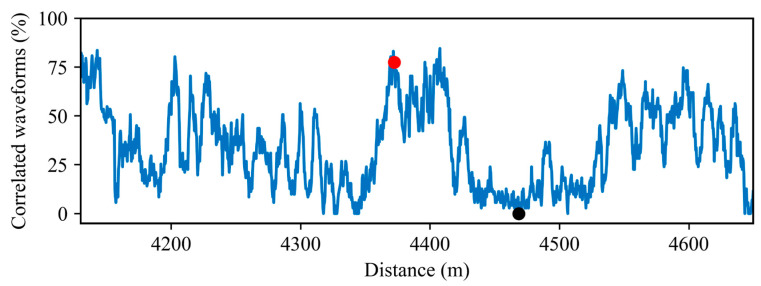
Correlated waveform percentage (blue line) for a 520 m long fiber section. z=4372 m, and z=4468 m are indicated by the red and black dots, respectively.

**Figure 9 sensors-23-06599-f009:**
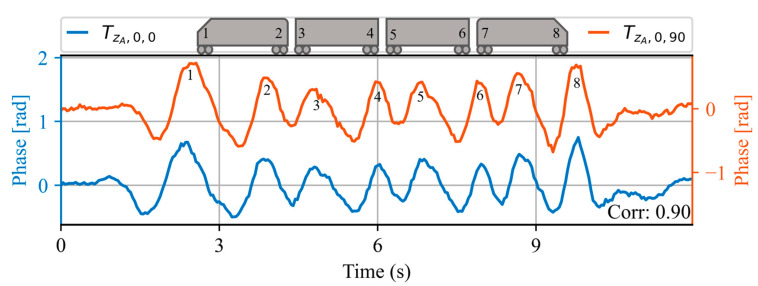
Acoustic waveforms of two trains at the same spatial location $Z_{A}$, measured within the same day. At the top of the figure, a 4-carriage train is illustrated, in which each of the bogies is numbered from 1 to 8. Note that the waveform contains 8 peaks corresponding to each of the bogies of the train. The correlation value obtained between the two waveforms is shown in the lower right corner.

**Figure 10 sensors-23-06599-f010:**
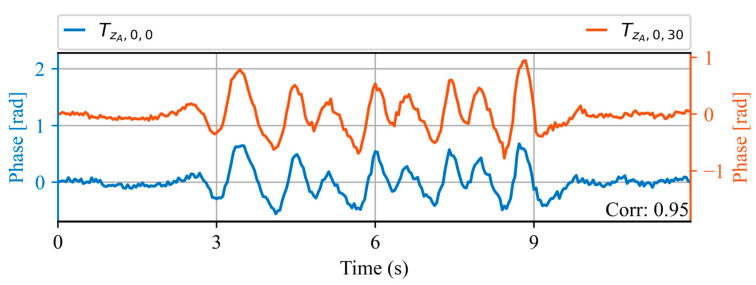
Acoustic waveforms of two trains at the same spatial location $Z_{A}$, measured within the same day. The correlation value obtained between the two waveforms is shown in the lower right corner.

**Figure 11 sensors-23-06599-f011:**
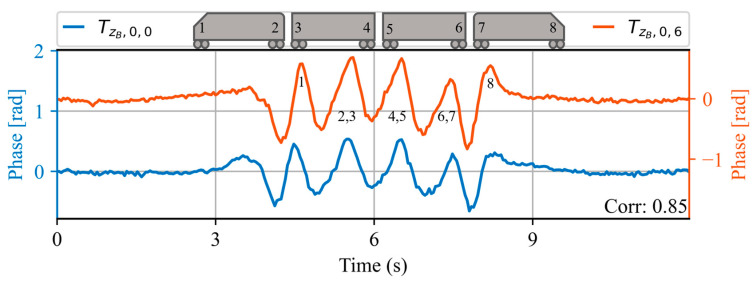
Acoustic waveforms of two trains at the same spatial location $Z_{B}$, measured within the same day. At the top of the figure, a 4-carriage train is illustrated, in which each of the bogies is numbered from 1 to 8. Note that the waveform contains 8 peaks corresponding to each of the bogies of the train. The correlation value obtained between the two waveforms is shown in the lower right corner.

**Figure 12 sensors-23-06599-f012:**
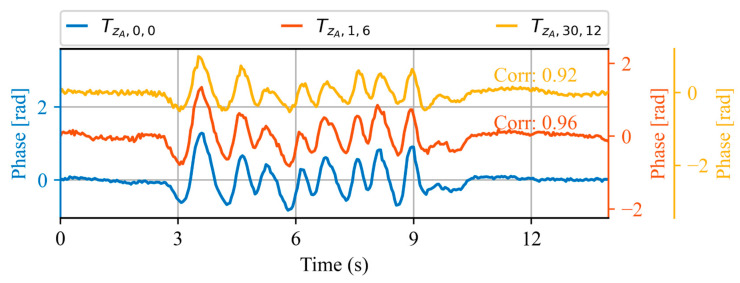
Acoustic waveforms of three trains at the same spatial location $Z_{A}$, measured at different days. The value of the correlation between the reference signal (blue) and the other signals (orange and yellow) is shown using the respective colors.

**Figure 13 sensors-23-06599-f013:**
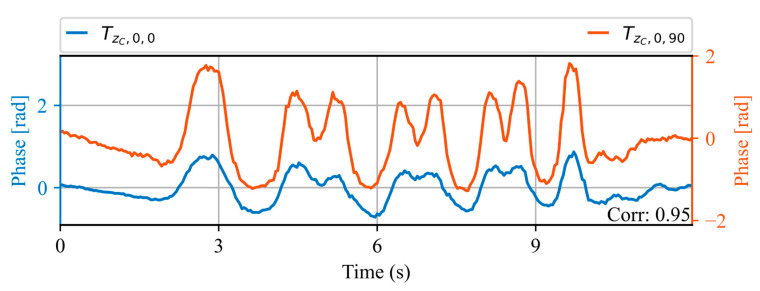
Acoustic waveforms of two trains at the same spatial location $Z_{C}$, measured within the same day. The correlation value obtained between the two waveforms is shown in the lower right corner.

## Data Availability

The data presented in this study are available by reasonable request from the corresponding author.

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
