# Peer review of "Finding Well-Coupled Optical Fiber Locations for Railway Monitoring Using Distributed Acoustic Sensing"

_sensors, 2023, doi:10.3390/s23146599_

Round 1

Reviewer 1 Report

The authors presented a rather interesting article on the processing and interpretation of a signal received from a distributed fiber optic sensor, which is used to monitor events on the railway track. The structure of the article and the style of presentation are good. The quality of the English language seems to me quite high. I am sure that this article will be of interest to readers of Sensors, since it contains interesting practical conclusions that allow one to interpret sensor events of this type. However, I have a few comments that I would like the authors to correct before publishing:

1. The introduction mentions quite a lot of previous work performed in solving similar problems using distributed acoustic sensors, but does not pay enough attention to various methods of signal preprocessing and postprocessing [1-3]. These stages of the algorithm play one of the most important roles in obtaining the result in this study.

2. From the point of view of the results repeatability, it would be advisable to present the cross-section of the sensor cable at least schematically. It would also be interesting to see a description, drawing or photo showing the topology of laying the cable in one of the locations of the track.

3. I would like to ask the authors to explain one thing: in line 207 you state that at the first stage, only signals up to 25 Hz were analyzed to identify low frequencies. A little later, in line 212, you claim that the data was downsampled at a frequency of 25 Hz. At the beginning of the manuscript, you mention the sampling theorem, which states that at this sampling rate, only high frequencies up to 12.5 Hz will be available. This confuses the reader a bit.

4.  In section 3.1 the authors claim that they obtain a "correlation function". Could you please explain why such a simple and already normalized criterion as the Pearson correlation coefficient was not used instead of the function? For two waveforms, it gives one number from minus one to one, which can be easily normalized, for example, within the range from 0 to 255 (for color interpretation) and you don't need to search for the maximum of the correlation function. Perhaps I did not understand the algorithm described by the authors well enough, but the answer to this question seems unobvious to me.

5. In line 231 you write about one of the processing routines: 'cleans the background noise or the image'. Could you please describe in more detail how this happens? Are MA/MD/FDDA filtration methods used? In one of the articles I suggested above [3], the authors show a serious increase in SNR using the simplest algorithms. This kind of signal cleaning can make the linear fitting more accurate.

6. Section 3 ends with figure 12. I would ask the authors to move this figure higher, immediately after the first mention.

[1] https://doi.org/10.1364/AO.58.004933

[2] https://doi.org/10.1364/OPTCON.460475

[3] http://dx.doi.org/10.3390/a16050217

[4] https://doi.org/10.3390/s22020413

Author Response

We thank the reviewer for taking the time to evaluate our article. Their comments have been of great value in helping us to improve the quality of our manuscript by supplementing the information and providing new references and figures. Modified sentences are highlighted in red in the revised version of the article.

Reviewer 2 Report

In the abstract, the purpose and the background should be further described to facilitate the readers to capture the importance content in this paper.

In the introduction, it is recommended to indicate that apart from the track, the overhead contact line should also be equipped with optical fiber sensors for health monitoring. Some literature is recommended to be included to indicate the necessity of monitoring this railway structure. For instance, [1] points out the necessity of detecting geometry faults of the overhead contact line, and [2] indicates the necessity of capturing the dynamic response of this structure subjected to wind load.

[1] Song, Y., et al. (2020). Contact wire irregularity stochastics and effect on high-speed railway pantograph-catenary interactions. IEEE Transactions on Instrumentation and Measurement, 69(10), 8196–8206. https://doi.org/10.1109/TIM.2020.2987457

[2] Fuchuan Duan, et al. (2023). Study on Aerodynamic Instability and Galloping Response of Rail Overhead Contact Line Based on Wind Tunnel Tests. IEEE Transactions on Vehicular Technology. https://doi.org/10.1109/TVT.2023.3243024

It is good to see that the shortcoming in previous research has been pointed out in the literature review. But it is recommended to summarise them before introducing the main work of this paper.

A more specific title is recommended to replace the ‘materials and methods’, which is too general but not informative.

Some outline texts are recommended to be added following the section title, which is necessary to give a glance for the readers.

As seen in the literature review that there are several mature techniques to tackle the problem. But the comparison with previous methods seems to be missing. Please comment on this issue. 

Author Response

(The authors gave the same response as above.)

Reviewer 3 Report

DAS has been used by authors on an over 30-year-old optical fiber installed alongside a railway track. A simple correlation-based method was proposed to automatically find spatial locations where temporal waveforms are repeatable, it is possible to extract useful information from trains crossing the railway track. The acquired signals could allow the estimation of the train position, speed, bogie count and possibly some parameter proportional to the weight of the train, experimental results indicate that they can be automatically discriminated.   

The paper is well written, it can be accepted after answering the following question:

1.      Regarding coherent fading noise problem or called “fading zone” in DAS, how did author handle this issue?

Author Response

(The authors gave the same response as above.)

Round 2

Reviewer 2 Report

All my comments have been well addressed. I recommend the publication of this paper.